# *Pseudomonas aeruginosa* Bloodstream Infections in SARS-CoV-2 Infected Patients: A Systematic Review

**DOI:** 10.3390/jcm12062252

**Published:** 2023-03-14

**Authors:** Marco Bongiovanni, Beatrice Barda

**Affiliations:** Division of Infectious Diseases, Ente Ospedaliero Cantonale, 6900 Lugano, Switzerland

**Keywords:** *Pseudomonas aeruginosa*, COVID-19, bloodstream infections, broad-spectrum antibiotic treatments, multi-drug resistant bacteria (MDR), antimicrobial stewardship

## Abstract

Bacterial co-infections increase the severity of respiratory viral infections and are frequent causes of mortality in COVID-19 infected subjects. During the COVID-19 period, especially at the beginning of the pandemic, an inappropriate use of broad-spectrum antibiotic treatments has been frequently described, mainly due to prolonged hospitalization, especially in intensive care unit departments, and the use of immune-suppressive treatments as steroids. This misuse has finally led to the occurrence of infections by multi-drug resistant (MDR) bacteria in hospitalized COVID-19 patients. Although different reports assessed the prevalence of Gram-negative infections in COVID-19 infected patients, scarce data are currently available on bloodstream infections caused by *Pseudomonas aeruginosa* in hospitalized COVID-19 patients. The aim of our systematic review is to describe data on this specific population and to discuss the possible implications that these co-infections could have in the management of COVID-19 pandemics in the future. We systematically analysed the current literature to find all the relevant articles that describe the occurrence of *P. aeruginosa* bloodstream infections in COVID-19 patients. We found 40 papers that described in detail *P. aeruginosa* HAIs-BSI in COVID-19 patients, including 756,067 patients overall. The occurrence of severe infections due to MDR bacteria had a significant impact in the management of hospitalized patients with COVID-19 infections, leading to a prolonged time of hospitalization and to a consequent increase in mortality. In the near future, the increased burden of MDR bacteria due to the COVID-19 pandemic might partially be reduced by maintaining the preventive measures of infection control implemented during the acute phase of the COVID-19 pandemic. Finally, we discuss how the COVID-19 pandemic changed the role of antimicrobial stewardship in healthcare settings, according to the isolation of MDR bacteria and how to restore on a large scale the optimization of antibiotic strategies in COVID-19 patients.

## 1. Introduction

On 31 December 2019, the WHO Western Pacific Regional Office was notified of reports of cases of “pneumonia of unknown cause” originating from Wuhan, China [1]. The disease, identified as being caused by a novel strain of coronavirus designated as severe acute respiratory syndrome coronavirus 2 (SARS-CoV-2), was named coronavirus disease 2019 (COVID-19). The rapid spread and the consequent rise of COVID-19 cases worldwide prompted the WHO to declare the disease a pandemic on 11 March 2020 [2]. Hence, COVID-19 became the largest WHO-recognized pandemic since the “Spanish ‘flu” in 1918–1920 [3], leading to more than 6 million people dying throughout the world.

At the very beginning of the pandemic, data on bacterial infections in COVID-19 patients were limited and anecdotic. In a retrospective analysis by Zhou et al. [4], sepsis was the most frequent complication in both survivors and non-survivors; however, no specific data were provided on the involved bacteria and outcomes. Similar findings were observed in another study from Wuhan: in this paper, patients with severe COVID-19 disease exhibited a significantly higher rate of bacterial co-infections compared with those with non-severe COVID-19 [5]. A wide range of positivity for co-infections or secondary infections has been reported across different studies, varying from 0.6% to 45% [6,7,8,9], underlying the need for larger studies including data on bacterial species and the site of infection.

Similarly, in the pre-COVID era, bacterial infections represented one of the leading cause of deaths. A large study investigated the global mortality associated with 33 bacterial pathogens in 2019, finding that *Staphylococcus aureus*, *Escherichia coli*, *Streptococcus pneumoniae*, *Klebsiella pneumoniae* and *Pseudomonas aeruginosa* were responsible for more than 50% of the overall cases of death [10]. In addition, the SARS-CoV-2 pandemic led to a dramatic increase in the prolonged time of hospitalizations and finally to a high rate of hospital-acquired infections (HAIs) [11,12,13,14]. The most frequent HAIs are ventilator-associated pneumonia (VAP), bloodstream infections (BSIs) and urinary tract infections (UTIs) that are often causes of sepsis and septic shock, especially in critically ill patients. In particular, *S. aureus* has been identified as the main pathogen in both HAIs and BSIs in COVID-19 individuals [15,16]. In particular, a study by Cusumano et al. clearly demonstrated that *S. aureus* infections are usually healthcare associated in COVID-19 subjects and have a mortality of 66.7% by day 30 after the first positive blood culture [16]. Furthermore, *Enterococcus* spp. colonization was reported to increase significantly in the gut microbiome of patients with severe COVID-19 infections [17]. As a consequence, *Enterococcus faecalis* and *Enterococcus faecium* have been identified as possible cause of BSI in individuals with COVID-19 [18,19]. These patients have usually a prolonged hospital stay, aggravated by intensive care unit (ICU) hospitalization and, hence, several HAIs can complicate their clinical outcome [12,20,21]. In COVID-19 patients, the risk of developing bacterial infections can be also exacerbated by prolonged immunosuppression due to the steroid treatment administered for lung involvement [7,22].

Several papers have reported a higher incidence of multi-drug resistant (MDR) bacteria in COVID-19 individuals [22,23,24,25,26,27,28,29,30], mainly associated with ICU hospitalization or an inappropriate use of antibiotic treatment, especially during the first pandemic wave [7,22]. One of the first reports on the pandemic assessed the prevalence of methicillin-resistant *S. aureus* (MRSA) in the respiratory cultures of patients with severe COVID-19 pneumonia. It reported that at day 28 after admission, MRSA prevalence increased from 0.6% to 5.7% [31]; in line with the above-mentioned data, the authors concluded that the occurrence of MRSA in COVID-19 related pneumonia is more likely to be an HAI or ventilator-associated complication rather than a community-acquired infection. In addition, MRSA has been frequently isolated in critically ill COVID-19 patients who developed VAP [32]. MRSA was not the only resistant bacteria with an increased prevalence during the pandemic; the spread of vancomycin-resistant enterococci (VRE) to COVID-19 patients in healthcare settings was demonstrated through the application of whole genome sequencing. In particular, samples from hospital surfaces and from both COVID-19 and non-COVID-19 patients were examined, finding a genetic correlation between the isolated bacterial strains, thus suggesting that contaminated surfaces played an important role in VRE nosocomial transmission during the COVID-19 pandemic [33].

Recently, an overall increase in different MDR bacteria in hospitalized patients with COVID-19 infection in Italy has been demonstrated [34]. In this study, the incidences of MDR infections in the first year of the pandemic was compared with that in the pre-COVID-19 period (2017–2019); in detail, a significant increase in the incidence of carbapenem-resistant *Enterobacteriacee* spp., VRE and carbapenem-resistant *Acinetobacter baumannii* was found in 2020, compared to the pre-COVID-19 era [34]. These data confirmed other reports from different countries [35,36,37,38], and represented an alert for the management of infective complications of COVID-19 infection, especially in those countries with a high prevalence of MDR bacteria.

Although different reports assessed the prevalence of Gram-negative infections in COVID-19 infected patients, scarce data are currently available on BSIs caused by *Pseudomonas aeruginosa* in hospitalized COVID-19 individuals. The aim of this review is to describe data on this specific population and to discuss the possible implications that these co-infections could have in the management of COVID-19 pandemics in the future.

## 2. Methods

We performed a systematic review on BSIs caused by *P. aeruginosa* in COVID-19 hospitalized patients. We searched PubMed, LitCovid, Web of Science and Embase for COVID-19 patients with a BSI caused by *P. aeruginosa*. The timeframe considered was from March 2020 until September 2022. We limited the research to human and studies written in English. We excluded papers that were written in languages other than English or that had no full paper available. MedRxiv was excluded because includes not peer-reviewed papers. We used as search terms “coronavirus infection” or “SARS coronavirus” or “severe acute respiratory syndrome” or “coronavirus” or “COVID-19” or “SARS-CoV-2” and “*Pseudomonas aeruginosa* infection” or “*Pseudomonas aeruginosa* bloodstream infection”, “*Pseudomonas aeruginosa* bacteraemia” or “*Pseudomonas aeruginosa* superinfection” or “*Pseudomonas aeruginosa* concomitant infection”. The strategy was focused on COVID-19 confirmed patients; the *P. aeruginosa* BSI must have been confirmed by a positive blood culture. Editorials, letters to the editor and short communications were excluded from the review. Studies in which none of the patients had COVID-19 or a BSI by *P. aeruginosa* were also excluded. Two independent reviewers (MB and BB) screened the abstracts of the identified studies and reviewed the full texts of those potentially eligible, with disagreements resolved by consensus. Data were extrapolated and the study details recorded: the authors, study design, sample size, geographical distribution, demographics, microbiology, drug resistance and length to BSI diagnosis, relevant comorbidities of the patients when specified (i.e., immune suppression, cardiovascular, diabetes, cancer, auto-immune, renal, lung or liver diseases), risk factors or correlated factors. We conducted this systematic review in accordance with the Preferred Reporting Items for Systematic Reviews and Meta-Analyses (PRISMA) guidelines (see also Appendix A for details) [39].

## 3. Results

We found a total of 12,800 papers on bloodstream infections in COVID-19 patients; 467 papers mentioned *P. aeruginosa* in COVID-19. Of these, 40 papers described in details HAIs-BSI in COVID-19 patients (Figure 1). Table 1 summarizes the studies selected according to the country where the research was conducted and to the microbiological methods used to identify *P. aeruginosa* infections, when available. From the selected papers 15/40 (37.5%) reported data from ICU isolates, 12/40 (30%) from medicine ward isolates, 4/40 (10%) from ICUs and medicine ward isolates and 9/40 (22.5%) from unspecified wards. The majority of studies (28/40, 70%) were retrospective observational studies, prospective observational studies were *n* = 3, systematic review *n* = 3, retrospective case-control studies *n* = 2, case reports *n* = 1, point-prevalence surveys *n* = 1, retrospective case series *n* = 1 and surveillance studies *n* = 1. Five studies included fewer than 100 patients and only 8 studies did not report the exact number of *P. aeruginosa* isolates on blood cultures.

Bacteria susceptibility was interpreted according to EUCAST (the European Committee on Antimicrobial Susceptibility Testing) criteria [40]. The time to blood culture positivity for *P. aeruginosa* was in all cases above 2 days from admission, suggesting HAI [30,35,41,42,43,44,45,46,47,48,49,50,51,52,53]. Only Yu et al. reported a mean time to positivity of 22.8 h [41] and Russel et al. found 11 out of 75 patients with *P. aeruginosa* BSI after fewer than 2 days [52].

One of the main topics during the first pandemic wave was the high number of BSIs observed compared to the pre-COVID period; as a consequence, different reports described a higher number of contaminated blood cultures in the same period [41,54,55]. For example, a study from Switzerland found 1099 contaminated blood cultures vs. 1616 ICU related BSIs, with a higher rate of false-positive blood cultures during the pandemic peaks [54]. The three considered periods in this study were the first pandemic wave, an interim period between May and October 2020 and a second pandemic wave until May 2021. The contamination rate was around 40% during the pandemic and 35% during the interim period. The authors suggested that the higher number of samples from the peripheral veins, reallocated staff who had not been not trained enough that was implied during the pandemic and the scarce accuracy in the aseptic techniques due to ward and ICU overcrowding were all possible factors associated with these results [54,56]. These findings might have a big impact on the duration of hospitalization and on the unneeded prescriptions of antibiotic treatment, finally leading to augmented hospitalization costs [56,57].

Different clinical conditions have been proposed as possible explanations for the increased rate of BSIs during the COVID-19 pandemic, such as the prolonged hospitalization, especially during the first pandemic wave [58,59], the use of high-dose systemic corticosteroids treatment for acute respiratory distress in the ICU [20,29,42], the administration of immune-modulator treatment such as tocilizumab or baricitinib [29], the mechanical ventilation itself [29,43] and the worse clinical presentation at ICU admission (higher SOFA and APACHE II score) [30,42].

Sturm et al. evaluated the impact of the pandemic on community and hospital acquired BSIs in a multi-state healthcare system in United States. The authors did not observe significant differences between the pandemic and pre-pandemic period according to the rate of *P. aeruginosa* BSIs; however, during the pandemic, COVID-19 infected individuals had a reduced incidence of *P. aeruginosa* BSIs compared to non-COVID-19. Opposite results were found when HAIs were considered: although no significant differences were observed during the pandemic and pre-pandemic period, the rate of *P. aeruginosa* BSIs was approximately five times higher in COVID-19 subjects compared to non-COVID. In addition, Sturm et al. attributed the increase in HAIs during the COVID-19 period to the severity of COVID-19 illness, the immunosuppressant therapy prescribed and the vast exposure to antibiotics. Interestingly, it was observed that the routine measures taken to mitigate the HAIs such as universal decolonization, alterations of the central and peripheral line care and reduced compliance to hospital hygiene protocols were disrupted [60]. Other studies compared the pre-COVID-19 and COVID-19 incidence of BSIs with discordant results. In particular, Gaspari et al. [51] reported BSIs in 4/173 COVID-19 patients vs. 7/132 non-COVID-19 patients. Lai et al. [7] showed a slightly higher number of BSIs by *P. aeruginosa* among non-COVID-19 patients compared to COVID-19 ones (1440 vs. 1240). Gaspari et al. focused their research on ESKAPE pathogens (*Enterococcus faecium*, *Staphylococcus aureus*, *Klebsiella pneumoniae*, *Acinetobacter baumannii*, *Pseudomonas aeruginosa* and *Enterobacter* spp.) and their involvement in HAIs. The retrospective data of patients admitted to ICU during pre-COVID-19 and COVID-19 periods were collected and analysed with a special focus on HA-BSIs according to the ESKAPE pathogens. They evaluated 305 patients who spent more than 48 h in ICUs (173 COVID-19 and 132 non-COVID-19 patients), observing 26.5% vs. 13.3% of ESKAPE BSI in non-COVID-19 and COVID-19 patients, respectively [51]. Torecillas et al. in a retrospective trial assessed the impact of antimicrobial therapy during the pandemic compared to the pre-pandemic period, with a specific focus on BSI in ICU. They reported an overall decrease in some MDR isolates, such as ESBL or CPE and MRSA but, on the other hand, an increased rate of *P. aeruginosa* resistant strains with an RR 4.2 (95% CI: 1.4–12.8) [61]. Opposite to these results, LeGlass et al. gave a snapshot from the South of France during the pandemic; COVID-19 patients were compared with non-COVID-19 ones and patients infected with influenza viruses. A higher number of co-infections among COVID-19 patients with a 2.9% prevalence, compared to 1.6% of non-COVID-19 patients and 1.0% of the influenza-infected ones, was reported. The COVID-19 patients had a higher prevalence of HA-BSIs, overall 72.2% vs. 48.2% and 39.5% in non-COVID-19 and influenza patient controls, respectively. Among these, *P. aeruginosa* BSI was significantly more common in non-COVID-19 patients than in controls (*p* = 0.002), both in medicine wards and in ICUs, and it was more common in the respiratory samples than in blood ones [62]. In addition, Meschiari et al. noticed a reduction in the prevalence of *P. aeruginosa* susceptible strains during the pandemic [63].

Patients’ comorbidities were also taken into account, together with other COVID-19 related risk factors as possible predictors of *P. aeruginosa* BSIs. Buetti et al. found that cardiovascular and metabolic diseases were the most common comorbidities (60% cumulatively), followed by respiratory diseases and malignancies (17% respectively) [30]. Mantzarlis et al. did not find significant differences comparing COVID-19 patients with and without BSI [64]. In both groups, metabolic and cardiovascular diseases were the most common comorbidities [30,58,64]. Similarly, Posteraro et al. did not find any correlation between fatal COVID-19 and other concomitant diseases [51]. A slightly higher rate of BSIs by *P. aeruginosa*, though not statistically significant, was mainly observed in patients with chronic lung disease [58].

The increase in MDR pathogens during the pandemic is actually highly debated [65]. The increase in resistance might be due to hospital overcrowding, inexperienced applied personnel, the disappearance of the antimicrobial stewardship, less accuracy in laboratory pre-analytic attention to avoid contamination and the surveillance of diagnostic tests to detect resistant germs. In support of this thesis, an increase in the incidence of antimicrobial resistance in critically ill COVID-19 patients was described, especially for *A. baumanii*, *S. pyogenes* and *H. influenzae* despite the increased use of control measures [7,61]. One of the possible explanations might be a misuse or an inappropriate use of antibiotics during the first phase of the pandemic, together with poor adherence to the guidelines and antimicrobial stewardship protocols. Another risk factor for developing MDR *P. aeruginosa* BSIs was represented by the patients colonized by MDR bacteria, or with a previous exposure to broad-spectrum antibiotics [58]; the occurrence of MDR isolates was associated with the use of vancomycin, ceftriaxone, piperacillin/tazobactam and carbapenems [20,60]. On the other hand, there are also reasons against the increase in MDR pathogens, such as thorough attention to infection control measures adopted during the pandemic, hand hygiene, the use of personal protective equipment and tools to decontaminate surfaces and air [65]. In addition, few data on MDR bacteria included *P. aeruginosa* strains, which apparently had less resistance than predicted [44]. Similarly, Palanisamy et al. described that 8.5% of BSIs were mainly due to *K. pneumonia* and *A. baumanii*, and only 4% of *P. aeruginosa* were without main resistance [43]. An Italian review found that patients who underwent many cycles of prone–supine positions had a higher chance of developing MDR superinfections, mainly due to the extensive use of antibiotics during the pandemic [66]. Almost all the authors agree on the misuse of antibiotics during the pandemic was a crucial risk factor for the increased number of MDR bacteria. Grasselli et al. described the condition of eight Italian hospitals, in which 35% of infections were due to the MDR bacteria. Almost 70% of these patients were receiving broad-spectrum empiric antibiotics, which was demonstrated to be an independent risk factor for the development of HAIs [49]. A similar trend was observed across Europe, where an increase in antibiotic consumption together with a decrease in prescription accuracy was registered. Overall, the beginning of the pandemic was the period most affected by this phenomenon, also aggravated by the abuse of azithromycin, which was initially recommended for managing COVID-19 [63]. During the first 2 months of the pandemic, a dramatic increase in the prescription of azithromycin, ceftaroline, linezolid and echinocandins was found; for carbapenems, daptomicin and novel cephalosporin/β-lactamase inhibitors a steady increase in the prescriptions was also observed during the last few years, but this trend was not confirmed during COVID-19 time. Of note, vancomycin prescriptions dropped as low as almost zero during the timeframe considered [67]. On the contrary, Mentzatlis et al. did not find any significant correlation between the risk of developing an HA-BSI and the administration of antibiotics [64]; Gago et al., on the other hand, found a significant correlation between the prescriptions of vancomycin, carbapenems and piperacillin/tazobactam in COVID-19 patients and a higher odds ratio of developing an HA-BSI [20].

### Future Perspectives and Antimicrobial Stewardship

The excessive empiric use of broad-spectrum antibiotics observed in COVID-19 patients, especially during the first wave of the pandemic, is heightening concern that this overuse will exacerbate the problem of antimicrobial resistance in microorganisms of clinical significance in the near future [68,69,70]. In both critical and non-critical settings, the use of broad-spectrum antibiotic treatment was prescribed in up to 72% of COVID-19 patients [6]. These findings were confirmed by a large meta-analysis that described 3338 hospitalized and critical COVID-19 patients across 24 studies. This study revealed that >70% of COVID-19 patients were treated with antibiotics [71], despite the fact that bacterial co-infection and secondary infection rates were 3.5% and 14.3%, respectively. The authors stated that “there is currently insufficient evidence to support widespread empirical use of antibiotics in most hospitalized patients”. Overall, the rate of bacterial infection was higher in critically ill COVID-19 patients than in those hospitalized with milder symptoms. These data on a misuse of antibiotics were also confirmed in another meta-analysis that found 14% of bacteria co-infections in COVID-19 patients in ICUs and 4% in those hospitalized in other departments [72]. All these considerations make the role of antimicrobial stewardship mandatory in COVID-19 infections with different grades of severity. During the early phase of the pandemic, due to the overcrowding of the hospitals, the antimicrobial stewardship programmes were forced to be set aside in most hospitals. This led to inappropriate prescriptions of several broad-spectrum antibiotics that contributed to the selection of MDR bacteria in COVID-19 patients. The correct therapeutic approach in patients with suspected bacterial infections would be to select the treatment according to the isolated germ, if available, the local epidemiology and the patient’s clinical conditions and clinical history; if the infection is not confirmed, an early discontinuation of the antibiotic is advisable. The WHO guidelines strengthened the role of antimicrobial stewardship in COVID-19 patients and advised against the use of antibiotic therapy or prophylaxis in patients with suspected or confirmed mild or moderate COVID-19, unless there is a clinical suspicion of a bacterial infection [73]. Furthermore, the WHO recommends that antimicrobial therapy should be assessed daily for possible and prompt de-escalation. In addition, NICE guidelines confirm that the optimization of antibiotic use should be mandatory in COVID-19 patients, underlying the inappropriateness of antibiotics for prevention or treatment of COVID-19 and advising antibiotics only where microbiologically confirmed or in cases of a strong clinical suspicion [74]. In the current pandemic phase, with less severe COVID-19 infections, restoring and implementing the antimicrobial stewardship programmes should be of primary importance for all the hospitals managing COVID-19 patients.

## 4. Discussion

In the past 2 years, many papers and reviews have reported data on HA-BSIs in COVID-19 patients, but to our knowledge, this is the first one that focuses on *P. aeruginosa* BSIs. Digging into the literature, we noticed that *P. aeruginosa* BSIs overall increased among COVID-19 patients due to multiple factors. As learnt during the pandemic, severe COVID-19 cases are burdened by a higher mortality compared to non-COVID-19 patients; the infection involves different organs and often leads to multi-organ failure, although the pulmonary disease is mainly responsible for prolonged hospitalizations. Both internal medicine and ICU long hospitalizations put the patient at risk of developing nosocomial infections. Moreover, the prolonged use of steroids and immunomodulatory therapies plays a role in the immune depression of these patients, increasing the risk of HAIs. Among the drugs used for COVID-19, tocilizumab has showed a correlation with *P. aeruginosa* BSIs [75,76]. Peng et al. reported a decreased mortality among COVID-19 patients treated with tocilizumab, but a higher number of fungal infections [77]; Minihan et al. also noticed that among COVID-19 patients who received tocilizumab, the number of infections was double that of patients who did not receive it [76].

As mentioned above, the baseline clinical conditions of the patients are an important factor for the development of severe COVID-19 and, therefore, for superinfections. Patients with critical laboratory and clinical presentation had a higher probability of developing bacterial infections [30,48,54,64]. The use or, more precisely, the abuse of antibiotics in the first months of the pandemic has resulted in a rapid, wide spread of MDR bacteria [20,42,46]. Luckily, most COVID-19 patients were already isolated and MDR bacteria did not exponentially increase in hospital settings, but some serious concern is rising, with some authors reporting a high incidence of MDR *P. aeruginosa* in up to 50% of the isolates [47,51]. However, it is still unknown whether the maintenance of the intensified measures of the protection from infections, prevention and control measures beyond the COVID-19 pandemic could significantly improve the control of MDR bacterial infections in the hospital setting. Finally, we believe that valuable lessons can be drawn from the COVID-19 pandemic in terms of understanding the importance of infection control and antimicrobial stewardship in healthcare settings to reduce the burden of HAIs and antibiotic resistance in the future.

## 5. Conclusions

Scientific evidence underlines that patients with COVID-19 infection are at risk of acquiring HAIs. This risk is significantly associated with the severity of COVID-19 disease and the duration of hospitalization. Many of the bacterial HAIs in this population exhibit a resistance to different antibiotics, mainly secondary to the high empiric use of broad-spectrum antibiotics documented in COVID-19 individuals. Conversely, the implementation of the preventing measures to reduce the risk of nosocomial transmission of COVID-19 is linked to a reduced incidence of some bacterial HAIs, suggesting the need of maintaining these measures beyond the COVID-19 pandemic.

We believe that the COVID-19 pandemic made most healthcare providers realize the importance of antimicrobial stewardship and infection control practices to lower the burden of HAIs and, finally, antimicrobial resistance in the future.

## Figures and Tables

**Figure 1 jcm-12-02252-f001:**
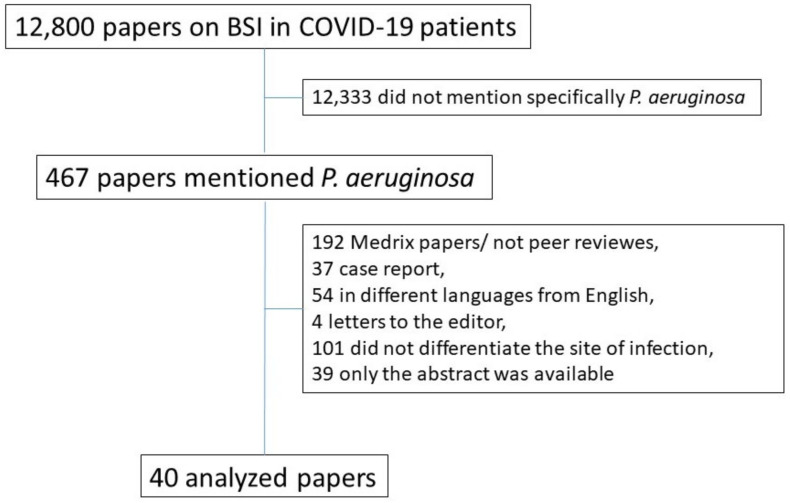
Flowchart of paper selection.

**Table 1 jcm-12-02252-t001:** Summary of the studies included in relation to country and the microbiological methods used to identify *Pseudomonas aeruginosa* infection.

	Author	Place	Microbiology Method
1	Søgaard	Switzerland	MALDI-TOF/PCR
2	Gago	USA	NA
3	Ventoulis	Greece	NA
4	Pasquini	Italy	NA
5	Naveenraj	India	VITEK-2
6	Adelman	USA	MALDI-TOF
7	Baiou	Qatar	MALDI-TOF
8	Bardi	Spain	NA
9	Buetti	Multicontinental	NA
10	Charlie	Canada	NA
11	Deiana	Italy, Sardinia	NA
12	Dezza	Italy, Rome	NA
13	Doubravská	Chzec republic	MALDI-TOF
14	Gaspari	Italy, Rome	MALDI biotyper and VITEK2
15	Garcia-Vidal	Spain	NA
16	Grasselli	Italy	NA
17	Hirabayashi	Japan	NA
18	Hughes	UK	MALDI; EUCAST
19	Kariyawasam	Multicentre	EUCAST susceptibility
20	Lai	Taiwan	NA
21	LeGlass	France	NA
22	Mantzarlis	Greece	VITEK2, EUCAST
23	Meschiari	Italy	NA
24	Montrucchio	Italy	MALDI-TOF
25	Nori	New York city	NA
26	Orsini	USA	NA
27	Posteraro	Italy	MALDI-TOF
28	Pourajam	Iran	API 20E (bioMerieux Marcy-l’Etoile, France
29	Ripa	Italy	NA
30	Rouze	Europe	Culture
31	Russell	Scotland	NA
32	Santos	Multicentre	NA
33	Scott	Missouri, USA	NA
34	Sepulveda	New York	NA
35	Sturm	USA	NA
36	Subhashree	India	VITEK2
37	Torrecillas	Spain	MALDI-TOF
38	Amarsy	France	EUCAST
39	Yu	Sweden	MALDI-TOF/EUCAST
40	Cuntro	Italy	MALDI-TOF/VITEK

## Data Availability

Not applicable.

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
