# Peer review of "Pseudomonas aeruginosa Bloodstream Infections in SARS-CoV-2 Infected Patients: A Systematic Review"

_jcm, 2023, doi:10.3390/jcm12062252_

Round 1
Reviewer 1 Report
Timely and important.
Well done, methods clearly explained.
Minor language check warranted.
Please embed in your paper and cite the following publication (not present at the moment)
Probably some comment on it in the f discussion is warranted
Global mortality associated with 33 bacterial pathogens in 2019: a systematic analysis for the Global Burden of Disease Study 2019
Collaborators expand- PMID: 36423648
- PMCID: PMC9763654
- DOI: 10.1016/S0140-6736(22)02185-7
Abstract
Background: Reducing the burden of death due to infection is an urgent global public health priority. Previous studies have estimated the number of deaths associated with drug-resistant infections and sepsis and found that infections remain a leading cause of death globally. Understanding the global burden of common bacterial pathogens (both susceptible and resistant to antimicrobials) is essential to identify the greatest threats to public health. To our knowledge, this is the first study to present global comprehensive estimates of deaths associated with 33 bacterial pathogens across 11 major infectious syndromes.
Methods: We estimated deaths associated with 33 bacterial genera or species across 11 infectious syndromes in 2019 using methods from the Global Burden of Diseases, Injuries, and Risk Factors Study (GBD) 2019, in addition to a subset of the input data described in the Global Burden of Antimicrobial Resistance 2019 study. This study included 343 million individual records or isolates covering 11 361 study-location-years. We used three modelling steps to estimate the number of deaths associated with each pathogen: deaths in which infection had a role, the fraction of deaths due to infection that are attributable to a given infectious syndrome, and the fraction of deaths due to an infectious syndrome that are attributable to a given pathogen. Estimates were produced for all ages and for males and females across 204 countries and territories in 2019. 95% uncertainty intervals (UIs) were calculated for final estimates of deaths and infections associated with the 33 bacterial pathogens following standard GBD methods by taking the 2·5th and 97·5th percentiles across 1000 posterior draws for each quantity of interest.
Findings: From an estimated 13·7 million (95% UI 10·9-17·1) infection-related deaths in 2019, there were 7·7 million deaths (5·7-10·2) associated with the 33 bacterial pathogens (both resistant and susceptible to antimicrobials) across the 11 infectious syndromes estimated in this study. We estimated deaths associated with the 33 bacterial pathogens to comprise 13·6% (10·2-18·1) of all global deaths and 56·2% (52·1-60·1) of all sepsis-related deaths in 2019. Five leading pathogens-Staphylococcus aureus, Escherichia coli, Streptococcus pneumoniae, Klebsiella pneumoniae, and Pseudomonas aeruginosa-were responsible for 54·9% (52·9-56·9) of deaths among the investigated bacteria. The deadliest infectious syndromes and pathogens varied by location and age. The age-standardised mortality rate associated with these bacterial pathogens was highest in the sub-Saharan Africa super-region, with 230 deaths (185-285) per 100 000 population, and lowest in the high-income super-region, with 52·2 deaths (37·4-71·5) per 100 000 population. S aureus was the leading bacterial cause of death in 135 countries and was also associated with the most deaths in individuals older than 15 years, globally. Among children younger than 5 years, S pneumoniae was the pathogen associated with the most deaths. In 2019, more than 6 million deaths occurred as a result of three bacterial infectious syndromes, with lower respiratory infections and bloodstream infections each causing more than 2 million deaths and peritoneal and intra-abdominal infections causing more than 1 million deaths.
Interpretation: The 33 bacterial pathogens that we investigated in this study are a substantial source of health loss globally, with considerable variation in their distribution across infectious syndromes and locations. Compared with GBD Level 3 underlying causes of death, deaths associated with these bacteria would rank as the second leading cause of death globally in 2019; hence, they should be considered an urgent priority for intervention within the global health community. Strategies to address the burden of bacterial infections include infection prevention, optimised use of antibiotics, improved capacity for microbiological analysis, vaccine development, and improved and more pervasive use of available vaccines. These estimates can be used to help set priorities for vaccine need, demand, and development.
Author Response
We want to thank the reviewer for the comments. We included the reference suggested in the introduction section
Reviewer 2 Report
This article summarizes literature reports on pseudomonas aeruginosa bloodstream infections in SARS- 2 CoV-2 infected patients, elaborating on laboratory culture findings, clinical conditions, comorbidities and most importantly, antibiotics use. I have the following suggestions for improvements.
1. The abstract has an imbalanced portion of background rather than describing the scope of the study. Suggest to add brief introduction of study method and major points in results.
2. The last sentence of the abstract does not read right.
3. In the method section, list number of publications with detailed comorbidity data.
Author Response
We thank the reviewer for the comments
Here are the answers required:
1) We add a brief sentence adding some of the results in the abstract
2) We believe that now you can read the last sentences (probably this was due to format problems)
3) We better specify in the methods section the co-morbidities as suggested; in addition, we also implemented the methods used to perform the review
Reviewer 3 Report
1) This article attempt to be a systematic review, but the methodology did not mention if the authors employ any systematic methodology or guidelines.
2) The authors should employ guidelines like PRISMA (https://www.prisma-statement.org/?AspxAutoDetectCookieSupport=1) or Cochrane methodology (https://training.cochrane.org/handbook/current).
3) The authors did not elaborate a figure were they explain the total number of articles founded, the articules eliminated and employed.
4) Despite that this article was written as an opinion article in their majority, some partes looks like a systematic review article.
5) With the actual published evidence and with the references listed by the author, is possible to elaborate a systematic review or even better, a meta-analysis.
Author Response
We want to thank the reviewer for the comments
These are our answers
1) We specified the methodology employed
2) We used PRISMA as guidelines and we added the relative reference
3) We included a figure as suggested
4) Our article is a systematic review. We better specified it in the methods section
5) See comment 4
Reviewer 4 Report
Title and abstract, ok. Perhaps it is more correct to refer to "muti-drug-resistant bacteria"Keywords. Some additional term is missing, eg “broad-spectrum antibiotic treatments” and/or “multi-drug-resistant bacteria (MDR)”
Introduction. It might be worth considering removing the paragraphs that mention infections by S. aureus, MRSA and Enterococcus spp, since the review deals with bacteraemias by P. aeruginosa.
Methodology. A question arises, why was the search limited to only the English language?
Results. The results presented in the first two paragraphs could perhaps appear in the form of graphs or tables.
Third and fourth paragraph allude to justify the results. It is impressive that they will have more room in the discussion section. Ditto the following paragraphs.
Doubt, why hasn't the subsection titled “Future prospects and antimicrobial stewardship”
been included in the discussion section?
Discussion. It may be a little short.
Tables and figures. They miss each other. Perhaps a PRISMA 2020 flowchart could be included.
Conclusions. Some sentences are missing by way of conclusion.
References. 75 updated citations. Review. In some of them there is the in uppercase and lowercase titles.
Author Response
We wish to thank the reviewer for the comments
Here are our answers
1) Actually our paper is mainly focused on P.aeruginosa bloodstream infections not on all multi-drug resistant bacteria (we believe that with the huge amount of paper at the moment available in literature is quite impossible to do a fully comprehensive systematic review)
2) We added as keywords what reviewer suggested
3) Thanks for this comment. However, we believe that in the introduction a more "general" part on other bacteria identified as complications of COVID-19 infection is appropriate and useful to introduce the main topic of our review. We used only paper in English because the most accreditated Journals use English as preferred language.
4) We agree with the reviewer, therefore we include a summary table
5) Our paper is a systematic review, therefore we discuss the findings mainly in the results section than in the discussion section. Further, the observation of a high rate of contaminated blood cultures at the beginning of pandemic was not the main topic of our review, so we did not focused on it in the discussion
6) We included a dedicated paragraph on antimicrobial stewardship and future perspectives because this is a hot topic and in our opinion it is worth to be discussed in a separate section
7) We are aware that discussion is quite short, but the main topics are discussed elsewhere in the text; as a consequence, we argue that repeat them also in the discussion section could have been pleonastic.
8) We included a figure as suggested, with the flowchart of paper selection; we also included as supplemental materiale the Prisma flowchart
9) We included a conclusion section
10) We revised the reference section
Round 2
Reviewer 3 Report
All the issue were resolved.